# Effects of Eccentric vs. Concentric Sports on Blood Muscular Damage Markers in Male Professional Players

**DOI:** 10.3390/biology11030343

**Published:** 2022-02-22

**Authors:** Alfredo Córdova-Martínez, Alberto Caballero-García, Hugo J. Bello, Daniel Perez-Valdecantos, Enrique Roche

**Affiliations:** 1Bioquímica, Biología Molecular y Fisiología, Facultad de Ciencias de la Salud, GIR: “Ejercicio físico y envejecimiento”, Universidad Valladolid, Campus Universitario “Los Pajaritos”, 42004 Soria, Spain; danielperezvaldecantos@gmail.com; 2Departamento de Anatomía y Radiología, Facultad de Ciencias de la Salud, GIR: “Ejercicio físico y envejecimiento”, Universidad Valladolid, Campus Universitario “Los Pajaritos”, 42004 Soria, Spain; albcab@ah.uva.es; 3Departamento Matemáticas, Escuela de Ingeniería de la Industria Forestal, Agronómica y de la Bioenergía, GIR: “Ejercicio físico y envejecimiento”, Universidad Valladolid, Campus Universitario “Los Pajaritos”, 42004 Soria, Spain; hjbello.wk@gmail.com; 4Instituto de Bioingeniería y Departamento de Biología Aplicada-Nutrición, Universidad Miguel Hernández, 03202 Elche, Spain; eroche@umh.es; 5Instituto de Investigación Sanitaria y Biomédica de Alicante (ISABIAL), 03010 Alicante, Spain; 6CIBER Fisiopatología de la Obesidad y Nutrición (CIBEROBN), Instituto de Salud Carlos III (ISCIII), 28029 Madrid, Spain

**Keywords:** basketball, cortisol, cycling, creatine kinase, testosterone, volleyball

## Abstract

**Simple Summary:**

Muscle eccentric contractions produce a higher degree of damage compared to concentric contractions. However, during sport practice (training and competition), eccentric as well as concentric actions appear at different levels. The presence of muscle-specific proteins in circulation is indicative of damage. The present report compares three sport disciplines: cycling, mainly concentric, volleyball, mainly eccentric in the legs and concentric in the arms, and basketball, mainly eccentric. The aim was to analyze the pattern of muscular injury blood markers in professional players in two moments of the season: after a training period and after a competition period. Results show that after a training period, muscle damage blood markers are higher in basketball and volleyball players, as expected due to their dominant eccentric component. However, during competition, these markers are higher in cyclists as a result of frequent eccentric actions. Therefore, the component eccentric–concentric is not defined exclusively by the sport discipline. The moment of the season (training vs. competition) has to be considered as well. This information could help sport professional to planify more specific training programs, preparations for competition, as well as post-exercise recovery.

**Abstract:**

Background: Repetitive eccentric contractions can lead to higher degree of damage compared to repetitive concentric contractions. However, this type of exercise does not reproduce the real situations during the season in competitive sport disciplines. Methods: We analyzed the pattern of muscle damage blood markers in male professionals from three disciplines: cycling (*n* = 18), mainly concentric, vs. basketball (*n* = 12) and volleyball (*n* = 14), both mainly eccentric. Circulating muscle markers were analyzed in two moments of the regular season: after a 20-day training (no competition) period (T1) and after a 20-day period of high demanding competition (T2). Results: Blood levels of creatine kinase and myoglobin (muscle markers) increased in all groups at T2 compared to T1 as a result of competition intensity. The lower increases were noticed in cyclists at the end of both periods. Testosterone levels decreased at T2 compared to T1 in all disciplines, with lower levels found in cyclists. However, cortisol plasma levels decreased in basketball and volleyball players at T2, but increased significantly in cyclists, suggesting a limited adaptation to the effort. Conclusions: The pattern of circulating muscle markers is different depending of the demanding efforts (training vs. competition) of each particular discipline.

## 1. Introduction

Differences in strength after eccentric and concentric training could be due to differences in neural activation [1]. The main consensus accepts that eccentric contractions can generate greater hypertrophy than concentric ones. This occurs because eccentric contractions induce rapid neuromuscular adaptations, increased IGF-1 (insulin-like growth factor-1) mRNA expression, proliferation of muscle satellite cells, and an increase in muscle protein synthesis [2]. However, sustained and intense eccentric exercise may increase the levels of muscle damage [3]. This is characterized by an increased presence of muscle proteins, such as creatine kinase (CK), lactate dehydrogenase (LDH) and myoglobin (Mb), in blood. At the same time, certain hormones increase as well, including cortisol, which activates catabolic processes and anti-anabolic actions by controlling protein turnover [4,5,6,7,8,9,10,11,12,13]. All these parameters indicate increased muscle damage, negatively affecting performance and delaying the time required for optimal recovery [7,10,13,14,15,16,17]. 

In normal physiological conditions, the plasma concentration of skeletal muscle proteins is low. However, loss of sarcolemma integrity may result after intense exercise execution or from a pathological situation. This leads to the progressive leakage of intracellular proteins, making them useful indirect indicators of muscle damage [18,19]. The information provided by the circulating muscle markers can be accompanied by a combination of functional methods that can help to quantify the magnitude of exercise-induced muscle damage (EIMD). 

EIMD is characterized by the destruction of skeletal muscle cells leading, as mentioned before, to the leakage of intracellular muscle proteins in serum [4]. Clinical features include loss of muscle strength, reduced range of motion, swelling, and delayed onset muscle soreness (DOMS) [20]. Muscle force-generating capacity allows an indirect quantification of EIMD and is determined as the maximal isometric voluntary contraction (MIVC) [5]. In this context, functional tests are useful because they estimate the status of muscle tissue and the period necessary for muscle degeneration and regeneration following EIMD [5]. In addition, severity of EIMD can be assessed by the information provided by associated events such as inflammation [21] and oxidative stress [22]. Symptoms following EIMD are useful as well to estimate the extent of muscle damage, including increased passive stiffness, localized soreness, and strength loss [3,5,23,24,25,26,27,28,29,30,31].

Regarding circulating muscle proteins, CK is the most accepted and specific muscle marker measured in the main part of EIMD studies [32,33]. Other markers to asses muscle damage include Mb, LDH, aspartate aminotransferase (AST) and alanine aminotransferase (ALT) [7,8,9,18]. However, the lesions of other tissues different from muscle also contribute to increase some of these blood markers. For instance, ALT can become elevated when other tissues (i.e., liver) are damaged, making it difficult to directly define whether the damage is centered only at the muscular tissue [11]. In addition, plasma CK concentrations are conditioned by many factors related to exercise, such as intensity, duration, training status or experience, as well as by genetics and gender, which determine the amount of muscle mass [5,7,9,18]. Altogether, the inter-individual variability of muscle damage is vast, and it cannot be evaluated by only taking into account one of the above mentioned mechanisms [4,10,34].

As indicated before, exercise types incorporating mainly eccentric (i.e., running) compared to concentric (i.e., cycling) muscle contractions result in bigger muscle damage [3]. In addition to the type of exercise, the time involved in exercise execution is other factor to consider. In this line, Overgaard et al. (2004) [35] reported that the degree of muscle enzyme leakage in endurance activities (eccentric or concentric) depends on the distance. The longer distance correlates positively to greater stresses, resulting in an increase in muscle tissue damage. This structural damage occurs due to the sustained demand in a situation of severe substrate depletion (metabolic damage vs. mechanical damage). 

Eccentric and concentric actions used to be studied under controlled laboratory conditions that do not mimic real sport situations. In real competition, exercise movements in all sport disciplines are a combination of concentric and eccentric muscle contractions. Several examples can illustrate this statement. For instance, in cycling, the predominant component is concentric, although eccentric actions appear during sprints and very demanding climbs. Therefore, cycling is a non-weight-bearing activity requiring negligible eccentric contractions contrary to running, another aerobic exercise, but involving repeated eccentric contractions [36,37,38,39]. In this line, the ability to generate eccentric high power outputs over a short period of time is an instrumental factor for cycling performance. This capacity allows the execution of demanding actions during competition, including to close a gap, break away from the pack, or to dispute a sprint. On the other hand, basketball is a team sport, challenging players to be supple during execution of strenuous actions [40]. The typical movement patterns, such as repeated jumping and sprints, are known to cause EIMD as a result of a combination of predominant eccentric actions with few but very intense concentric actions [41]. Therefore, basketball is mainly eccentric in the predominant action jumps to throw the ball. However, the end of the gesture with the arms has a predominant concentric component. Finally, volleyball involves mainly short bouts executed at high intensity, but with limited time to recover during the game. Volleyball players develop a lot of eccentric contractions in leg muscles during the vertical jump together with many ballistic movements, as well as a few concentric actions with the arms [42,43]. Therefore, volleyball has an evident eccentric component due to jumping and landing actions, but concentric when hitting the ball.

These particular sport disciplines (cycling, basketball and volleyball) were chosen in the present report to study the degree of muscular damage during real training/competition situations in each particular discipline, far from controlled interventions performed in laboratory conditions. Cycling (mainly concentric) implies to train a long time without muscular impact, but in a continuous way [14,15,16]. Basketball (mainly eccentric) demands supple actions from players, performing at the same time strenuous actions combined with active and passive recoveries [40,44]. Finally, volleyball (mainly eccentric) involves high-intensity and intermittent bouts performed at maximal accelerations, jumps and changes of direction to achieve successful receptions, blockages and ball spiking during a match [45,46]. Therefore, the aim of the present report was to analyze the pattern of muscular injury blood markers in professional players from three different sports with different participation of muscular contractions after a period of training and a period of competition during the regular season.

## 2. Materials and Methods

### 2.1. Participation in the Training and Competition Periods

Male professional players from basketball, volleyball and cycling participated in the study at 2 periods of the season: after a 20-day training period (T1) and after a 20-day period corresponding to the peak of their competitive seasons (T2). Cyclists (*n* = 18), basketball players (*n* = 12) and volleyball players (*n* = 14) were from different teams playing in the first categories of national and international level. This is a convenience sample, because it is difficult to find participants of high competition level participating in this type of intervention with the risk of disrupting the training/competition schedule. Statistical power of the sample is indicated in Section 2.5.

The training period (T1) corresponded to a moment of the season in which participants performed controlled exercises at lower intensity at the beginning but increasing it gradually. In this period, participants were working resistance routines: upper and lower part of the body for basketball and volleyball players, combined with technical work. The daily routine included two sessions: (a) a morning session with 30 min of jogging and 90 min of resistance training (strength work in the gym); and (b) an evening session, with a 3 h technical training to improve agility, speed, balance, coordination and ball driving. Cyclists were working resistance exercises only for the lower part of the body during 1 h in the gym. This routine was combined with endurance road cycling at 60–80% VO_2_max (4 h per day), accumulating around 600 km/week. Regarding resistance routines, all participants started at low intensity, but gradually increased, performing routines at 80–90% of RM (repetition maximum), doing several repetitions at maximal speed. In addition, cyclists trained climbing in mountain roads.

The competition period (T2) corresponded to days of high demanding performance combined with resting and training days. At T2, basketball and volleyball players competed in 2 demanding matches, the Wednesdays (European league) and Saturdays (National league). They performed an active recovery the day after the match and high demanding training sessions the rest of the days: resistance at 90% RM, doing several repetitions at maximal speed and combined with technical work (ball driving, jumps, coordination). The competition period (T2) for cyclists was a big cycling tour (“Vuelta a España”/Spanish Tour) with 19 days of very demanding performance over 3 weeks with very challenging stages (time trials, mountain climbing). Demanding stages were followed by 2 resting days with active recovering (1 day per week). The total distance covered by cyclists in this big race was around 3500 Km. A timeline of the different events of the study at T1 and T2 is indicated in Figure 1.

Training programs were obviously adjusted to the corresponding sport specialty and supervised by a multidisciplinary team. The training program was in line with the theory of periodization for endurance sports (cycling) and team sports (basketball and volleyball). Diet to cover daily energy expenditure was adjusted by the team dietitians depending of the energy expenditure during training or competition: 45–80 kcal/kg/day for cyclists [47], 25–35 kcal/kg/day for basketball players [48,49,50] and 50–80 kcal/kg/day for volleyball players [51,52].

### 2.2. Study Design 

The study was in line with Declaration of Helsinki. This study was approved by the local Ethics Committee of the University of Basque Country (CEISH/202R/2012). All participants provided written informed consent. Over the course of data collection, the participants were instructed to not smoke, consume alcohol, or take medications or supplements known to alter the muscular response to exercise. Participants were free from diseases and pathological processes that could alter the immune function. 

In the study performed with cyclists, we must follow the protocols of the “Union Cycliste Internationale” (UCI or International Cyclist Union) before starting the competition, confirming the absence of pathologies among participants. In addition, we must confirm that none of the cyclists were taking excluded drugs or medications, nor gave positive in the routinary doping control tests performed before and during competition according to the World Anti-Doping Agency (WADA) (www.ama-wada.org accessed on 20 January 2022). On the other hand, the study performed in basketball and volleyball players followed the same medical controls according WADA guidelines.

To control the correct health status of participants, two weeks before T1, they completed a medical questionnaire and a normal cardiopulmonary and electrocardiographic examination. Anthropometric characteristics were registered (Table 1). Direct measurements of maximal oxygen uptake (VO_2_max) were monitored by progressive stages of exercise (25 W min^−1^, up to maximal power) on a cycle ergometer (Monark 818, Bilthoven, Netherlands). Oxygen uptake was measured continuously using a Jaeger ergo-pneumotest (Eos-Compact, Jaeger, Wurzburg, Germany) (Table 1). Heart rate was monitored continuously by electrocardiogram. As a result, all participants presented an optimal health status.

### 2.3. Determination of Blood Markers for Muscle Damage

Participants were tutored to eat a non-fat dinner at least 10 h before the 2 blood extractions. Blood samples (around 15 mL) were obtained in Vacutainer tubes at 8:30 a.m. from the antecubital vein with the subject in fasting conditions and seated for 30 min in a comfortable supine position. Blood extraction and transportation were performed according the WADA guidelines. Blood samples were obtained at the end of T1 and T2. Blood was distributed in one tube with gel and clot activator (10 mL) to obtain serum. The rest was transferred to an ethylenediaminetetraacetic acid (EDTA) tube (3–5 mL) to obtain plasma. Immediately after filling, EDTA tubes containing blood were inverted 10 times and stored in a sealed box at 4 °C. Controlled temperature was assured during transportation. The specific tag (Libero Ti1, Elpro, Buchs, Switzerland) was used for temperature control. 

Muscle blood markers were determined by spectrophotometry in an autoanalyzer (Hitachi 917, Tokyo, Japan) as follows: total plasmatic proteins (TPP), LDH: Lactate dehydrogenase.creatinine and urea by colorimetry (Biuret); AST and ALT by enzymatic ultraviolet spectrophotometry; CK by an enzymatic color test; and Mb by immunoturbidometry. Serum cortisol levels were determined with an enzyme-linked fluorescent assay technique, a combination of ELISA linked to a fluorescence final lecture, with the aid of a multiparametric analyzer (Minividas, Biomerieux, Marcy l’Etoile, France). Testosterone levels were determined by ELISA (DRG testosterone ELISA kit, DRG Instruments GmbH, Marburg/Lahn, Germany). All analyses were performed in a hospital certified laboratory. 

### 2.4. Muscle Damage Assessment 

The physiological control of fatigue and muscle damage was assessed according to the results of the competitions. In this sense, all the athletes were at the performance level expected for their physical conditions and training status. The Borg subjective fatigue scale was completed by participants. As professionals in each of their specialties, they did not indicate limiting efforts, stating that “the demand was in accordance with what was expected for the competition held”.

### 2.5. Statistical Analysis 

Statistical analysis was performed using the programing languages Python and R, data were expressed using 95% confidence intervals for the mean. Sample sizes were 18, 14 and 12. The Cohen’s d value for all variables studied was greater than 0.9, which implies that for a significance level of 0.05, the sample size has a power greater than 0.8 for *n* = 18 and 0.7 for samples sizes of *n* = 14 and 12. The power means that the probability that the test correctly rejects the null hypothesis if the alternative hypothesis is true. After checking the normal distribution using the Shapiro–Wilk test, a two-way repeated measure analysis of variance (ANOVA) was carried out by Greenhouse–Geisser test to check the existence of an interaction effect (S×T) between the sport type (S) and the season training/competition time (T). Boxplots showing the interquartile range and the differences between groups were used. A linear discriminant analysis was carried out using the Python package scikit-learn as a form of dimensionality reduction to obtain a measurement of the separation and differences between the study groups. Finally, bivariate correlations between circulating proteins were tested using the Pearson rank order correlation test. A value of *p* < 0.05 was considered as significant.

## 3. Results

A linear discriminant analysis was performed using CK as the main variable related to muscular damage observed for the groups of participants (cycling, basketball and volleyball). Figure 2 shows the plot of the first 2 linear discriminant functions (sport discipline and training/competition period). In the case of cyclists, the plot shows that the discriminant analysis projection is able to capture differences regarding the considered variables, not only between cycling and the rest of sports, but also between the group of cyclists at T1 and at T2 (which are positioned in different quadrants). The remaining groups (basket and volley at T1 and T2), are plotted in the same cluster, pointing out that the discriminant analysis performed does not underline the differences between these groups. 

Table 2 shows the effects of the type of sport on TPP, plasma urea and creatinine levels and the activity of LDH at T1 and at T2. TPP levels in plasma were significantly influenced by the type of sport. Basketball and volleyball participants had significantly higher TPP levels in plasma than cyclists, both at the end of T1 and T2. In this context, TPP levels decreased in cyclists at T2 compared to T1, and compared to levels in basketball and volleyball players at the end of T1 and T2. Creatinine showed significant differences regarding the type of sport, with higher values in basketball and volleyball players compared to cyclists at T1 and T2. No changes during the studied period (T1 vs. T2) were observed into the different sport disciplines for creatinine. Basketball and volleyball players presented significant higher LDH activities in plasma than cyclists. However, LDH values reached at T2 in cyclists were similar to the LDH values of basketball and volleyball players. Urea levels maintained the initial values during the studied period for all sport disciplines.

Figure 3, Figure 4 and Figure 5 show the pattern of the circulating parameters (CK, Mb, AST, ALT, testosterone and cortisol) at T1 and T2. The pattern was similar in all groups displaying a tendency to increase (T1 vs. T2) for CK, Mb (Figure 3), AST and ALT (Figure 4). The exception was ALT in basketball players, displaying a tendency to decrease in T2 (Figure 4). Although the pattern of changes was similar (increases in T2 compared to T1), the levels of CK in cyclists were lower compared to basketball and volleyball players. Finally, testosterone decreases at T2 compared to T1 in the three sport disciplines, although the levels in cyclists are the lower (Figure 5). Regarding cortisol, it decreases at T2 compared to T1 in basketball and volleyball players, meanwhile this pattern is inverted in cyclists, increasing at T2 compared to T1 (Figure 5).

Quantification of all these parameters is shown in Table 3. CK and AST increased into each sport discipline at T2 compared to T1, being significant only in cyclists and basketball players. Meanwhile, the AST values of volleyball players were significantly lower compared to cyclists at T1 and T2. Mb increased significantly only in cyclist at T2. In addition, the Mb values at T1 in basketball and volleyball players were significantly higher compared to cyclists. Testosterone values decreased significantly at T2 compared to T1 in cyclists and volleyball players and tended to decrease in basketball players. Moreover, testosterone values at T1 in basketball and volleyball players were significantly higher compared to cyclists at T1. On the other hand, cortisol values increased significantly at T2 compared to T1 in cyclists, but decreased significantly in basketball players or displayed a tendency to decrease in volleyball players. In addition, the cortisol values of basketball players at T1 were significantly higher compared to cyclists at T1. Finally, ALT (mainly hepatic) was the only parameter that did not display significant differences regarding sport discipline nor the moment of the study. 

Table 4 shows the differences expressed as percentage comparing T1 vs. T2, for CK, Mb, AST and ALT in all sport disciplines. The greater change was observed in cyclists (mainly concentric sport) for all variables compared to the other disciplines. The lower changes were observed in volleyball players. Figure 6 shows the correlations between the blood markers of muscle damage and cortisol and testosterone. A strong correlation exists between CK and AST. However, cortisol and testosterone only have a mild correlation between them, with no correlation to the rest of protein blood markers. Finally, a strong positive correlation exists between AST and ALT. 

## 4. Discussion

This study compares the pattern of the different EIMD markers in two predominantly eccentric sports (basketball and volleyball) to a concentric one (cycling). The most relevant finding is that EIMD is greater in cycling than in volleyball and basketball, in opposition to the accepted statement that eccentric sports produce greater muscle damage. This is true considering only repetitive contractions under controlled conditions. However, the laboratory tests do not mimic the variety of situations that occur in a real competition. In this context, only during the T1 period the rule that eccentric disciplines produce more EIMD is accomplished. T1 corresponds to a training moment of the season where controlled movements and actions are executed. However, during competition (T2), the greater change is wider in the concentric discipline (cycling), as shown in Table 4. 

In addition, blood markers of muscle damage are not different between T1 and T2 in eccentric disciplines such as basketball and volleyball (Figure 2). A likely interpretation might suggest that training or competition display similar demanding efforts in these eccentric disciplines. In both cases (basketball and volleyball), participants performed hard sessions of training combined at the same time with highly demanding competitions. However, in cycling, these variables are positioned in a different quadrant. This suggests that blood markers during a training period differ from the competition period in cyclists. Road cycling is characterized by several competitions each season. Each competition consists of successive races (e.g., 2, 8 or 22 days) lasting around 4 h each day. During competition, cyclists undergo short recovery periods between stages (e.g., 24 h or less), which could impair subsequent performance. Particularly, in a big race such as “Vuelta a España”, the recovery is very short (less than 12 h). This particular situation favors a plasmatic increase of muscular proteins, as observed in this report and in others [38,39]. 

It has been reported that EIMD impairs muscle strength and power, resulting in a high impact for performance in different sport disciplines. Muscle damage could correlate to the different intensities and duration at which training actions and competition are performed. This may explain the variability in muscular blood markers of EIMD into the same sport and comparing the different disciplines (Figure 3, Figure 4 and Figure 5). We hypothesize that many particular aspects of each particular discipline could explain this variability. These variables include the role of each player in the team, the position of the player during the game or the time played during the competition (titulars vs. reserves), among others (see limitations at the end of this section). For this reason, the data from Table 3 have to be interpreted with caution because all participants did not undergo the same match/race situations, particularly in T2. Unfortunately, we could not homogenize all these variables without disturbing the season strategy of each professional team. 

Moreover, eccentric actions involve a lower metabolic demand on muscle tissue compared to concentric actions, but they require a greater mechanical demand [51,52]. However, both eccentric and concentric actions have been shown to be instrumental for optimal gains in muscular strength in response to resistance training [53,54,55]. In these cases, eccentric and concentric exercises produce sarcolemma disruption, as a result of increased proteolytic activity, and inflammatory response, which are manifested by plasmatic increases in muscle blood markers, such as CK [5,8,10,11,19,33]. Therefore, muscle damage can affect training performance and muscle adaptations [56]. 

In the current study, we have observed high levels (some of them are significant) in all muscular blood markers (CK, Mb, AST) at T1 in volleyball and basketball players compared to cyclists. Strength sessions are usual at T1 in basketball and volleyball players, explaining the high levels at the end of this period compared to cyclists. This could be likely due to the lower eccentric contraction component during training (T1) in cyclists. Training sessions for basketball and volleyball consist of multiple sprints and jumping exercises (plyometrics-eccentric). In this context, a good training planning (as it occurs in the teams studied) results in a positive change in performance, implementing the action efficiency and technical skills, as we have confirmed in a previous report [17]. However, at T2, the biggest change occurred in cyclists: 100% for CK vs. 80% and 11% in basketball and volleyball players, respectively. A similar tendency was observed for the rest of markers (Table 4). 

On the other hand, contacts against opponent players could contribute to EIMD during T2, particularly in basketball. These contacts are almost absent in cycling and volleyball. In this line, the contacts in basketball are not as intense as in other disciplines, such as soccer. Violent contacts in soccer can produce a muscle lesion and increase the circulating markers of EIMD. In basketball, these contacts are performed for position gain using the whole body. These actions result in modest EIMD compared to soccer contacts. In this context, Souglis et al. (2015) [57] have analyzed and compared the inflammation response and muscle damage in male elite players from 4 different disciplines: soccer, basketball, volleyball and handball. Muscle damage (determined by circulating increases of CK and LDH) showed an intermediate stress for basketball players. Results from this report [57] go in the same direction than in our study. 

Intense and sustained exercises compromise muscle integrity favoring the release to circulation of muscle proteins (CK, Mb and LDH), limiting performance [4]. Nevertheless, in our study, all participants maintained the maximal performance level along the studied period, despite blood increases in muscular markers. This observation suggests that certain levels of muscle damage do not seem to be a limiting step in performance. However, we can infer that when these levels are surpassed, the function and integrity of muscular tissue could be strongly compromised, increasing the risk of lesions. 

The underlying mechanisms are multifactorial and include neuroendocrinological and metabolic factors such as cortisol release [2,4,7,9,10,14,15,16,30,37]. Brancaccio et al. (2007 and 2008) [9,18] indicated that metabolically exhausted muscle exhibit compromises cell membrane integrity with a subsequent increase in cytosolic free calcium ions, promoting the activation of voltage dependent potassium channels. The same authors [9,18] show that the local tissue damage can be provoked by sarcomere deterioration from Z-band disruption, being CK a marker for muscle necrosis. Otherwise said, if exercise intensity goes from mild to moderate, muscular tissue is stressed, but with no marked changes in membrane permeability. When the exercise intensity exceeds a certain range, membrane permeability is affected and muscle proteins are released. Therefore, the limiting point is the resistance of myocyte to the mechanical and physiological stress [5]. 

In addition, sustained and intense endurance training can cause increases in the basal levels of cortisol (determined early in the morning) [58]. The levels of cortisol seem to correlate to the duration of workload rather than work intensity [59]. During the 3-week cycling competition (“Vuelta a España”), the metabolic homeostasis of participants seems to undergo a high stress. In this context, serum cortisol levels are indicative of accumulated stress in response to the high intensity during the race [13,17,58,59,60,61]. Higher intensity and longer duration correlate to higher circulating cortisol levels. Moreover, when the relationship training/recovery is optimally modulated, cortisol levels come back to basal levels within 24 h post-exercise [60,61]. In this context, Hough et al. (2013) [34] observed higher cortisol values after 11 days of extreme and high intense cycling training period, indicating high levels of stress and suboptimal recovery. In our opinion, the changes that we observe in cortisol and testosterone can be related to an accumulated stress, but with a correct tissue regeneration. In our study, despite being carried out in a period (T2) of maximum intensity for the three sport disciplines, cortisol levels increased significantly in cyclists and decreased in the others. However, the levels were in the same range for the three disciplines. In addition, the decrease in cortisol observed for basketball and volleyball players might be due the optimal recovery periods after exercise. 

The adaptation to stress triggered by exercise seems to be a good mechanism to maintain an optimal competitive level. However, an excess of stress results in disruption of the neuroendocrine axis, contributing to fatigue [62]. According to results in competition, participants show a good performance in the present study, indicating that the previous training program (T1) allowed an optimal muscle recovery. In addition, testosterone decreased comparing T2 vs. T1 in the present report in all participants from the three disciplines. During T2, basketball and volleyball players use to have 3 days/week training with 1 recovery day (Figure 1). The rest of the days of the week, intensity is maximal due to competition. Blood marker levels indicate that the post-exercise recovery strategies are very well planned. Nevertheless, cyclists were studied during a 3-week demanding competition (at T2) together with shorted recovery periods, which is reflected in the cortisol increases. 

Our hypothesis is that the changes that we observed in cortisol and testosterone in the present study might be related to accumulated stress during competition together with an optimal tissue regeneration during the recovery periods. This implies a balance in anabolic/catabolic processes due to a good adaptation to training and competition. We hypothesize that the observed differences mainly concern the type, duration and intensity of training. An altered modulation of this peripheral response may indicate training stress, sustained fatigue and possible overreaching. This hypothesis needs to be tested in future research.

The presented data are valuable to the broader scientific community or other athletes and coaches from comparable sport disciplines. EIMD could be significant in concentric disciplines under high demanding situations such as competition. This point needs to be taken into account to planify training as well as recovery sessions. Finally, and as mentioned before, particular aspects of each particular discipline (role/position of the player, time played) could be considered as limitations of the study. An additional limitation of the study is that participants correspond only to male players. It could be expected that the results would differ in a female population. In this context, hormonal cycles can condition the pattern of EIMD blood markers. Another limitation of the study is the number of athletes in each group (low *n*). Obtaining participants for this type of study is complicated because there are very few high-level professional athletes. A further limitation concerns obtaining blood samples or perform physiological measurements after each competition. Blood extractions were performed only at two moments of the season (at the end of T1 and T2). Therefore, more complete information could be obtained with more frequent blood extractions performed during the study period. Nevertheless, this schedule of extractions cannot be performed in professional players without disturbing from their goals. A final limitation concerns to the amount of muscle mass of the players in the different disciplines. Basketball and volleyball players are taller and stronger than cyclists, because they have more muscle mass. Therefore, the muscular impact of competition (observed in T2) is greater in cyclists, perhaps because the eccentric component in their regular training is very low, but high and demanding when competing, due to the changes in rhythm and sprints they perform. All these limitations need to be addressed in the design of the future research.

## 5. Conclusions

In conclusion, the present report comparing dominant concentric vs. dominant eccentric disciplines, shows that the circulating level of muscle markers derived from exercise-induced muscle damage is higher in basketball and volleyball players during a training period. However, exercise-induced muscle damage is significantly higher in cyclists due to competition.

## Figures and Tables

**Figure 1 biology-11-00343-f001:**
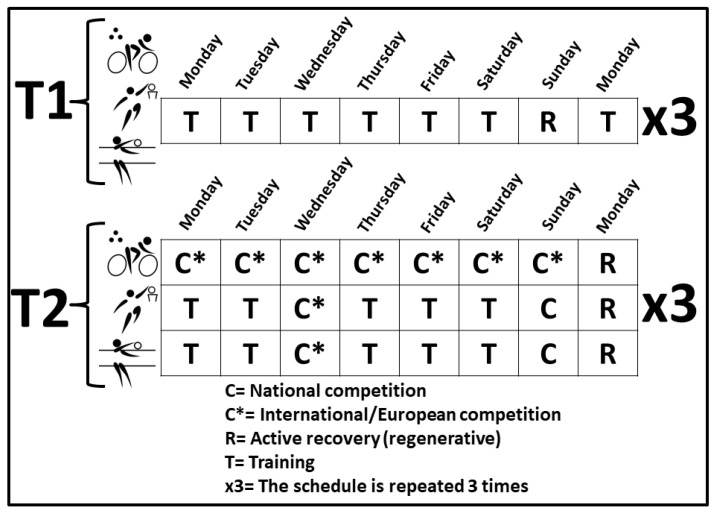
Timeline of events during the study in the 3 sport disciplines. Matches for basketball and volleyball are scheduled according to the coordinated program made by all European Federations.

**Figure 2 biology-11-00343-f002:**
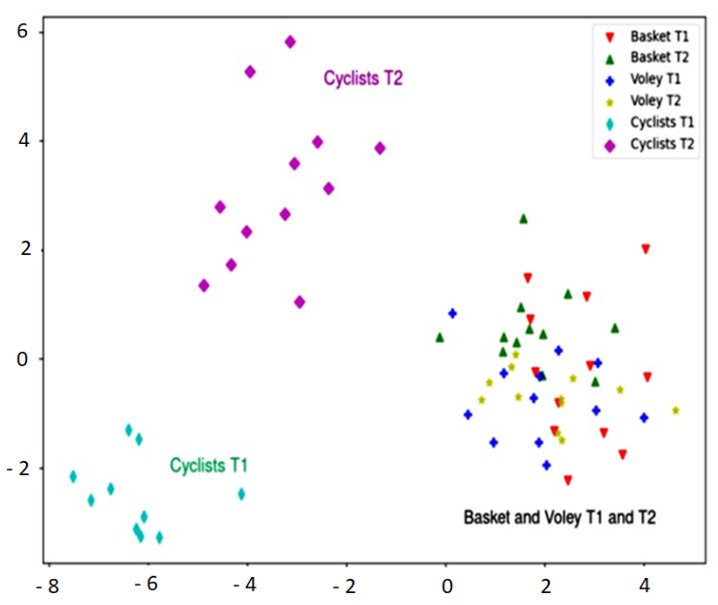
Linear discriminant analysis for cyclist, basketball and volleyball players at T1 and T2.

**Figure 3 biology-11-00343-f003:**
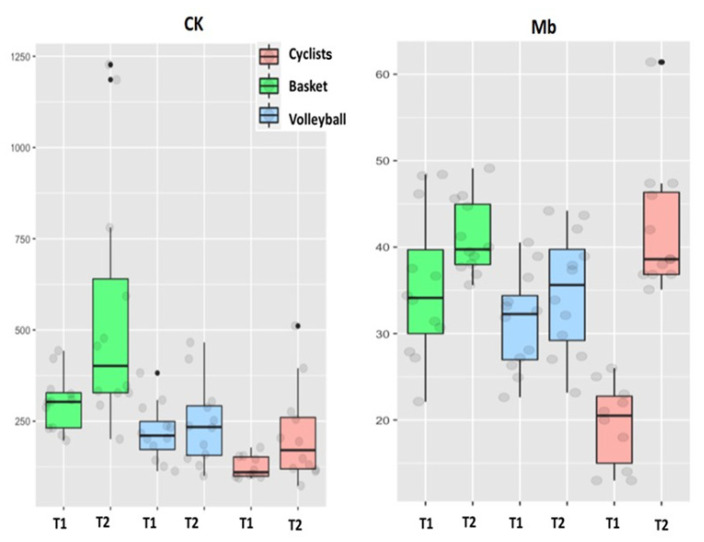
Creatine kinase (CK) (IU/L) and myoglobin (Mb) (ng/mL) plasma levels in cyclists, basketball and volleyball players at the end of T1 and T2.

**Figure 4 biology-11-00343-f004:**
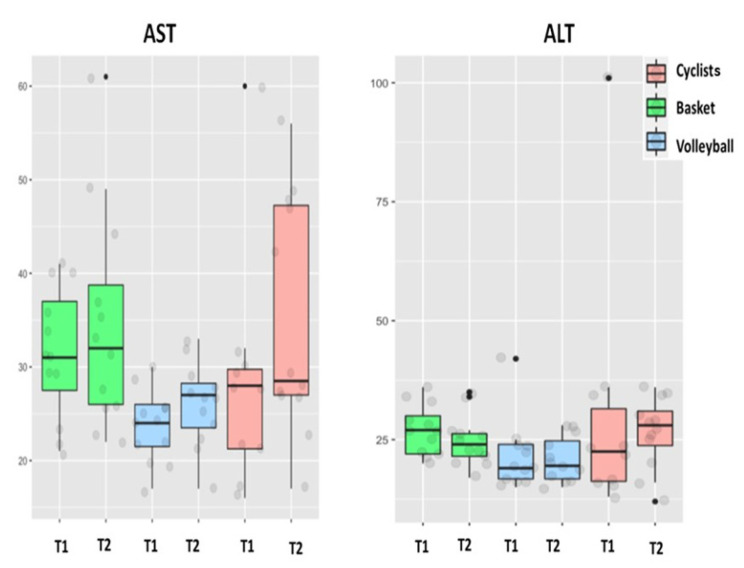
Aspartate aminotransferase (AST) and alanine aminotransferase (ALT) (IU/L) levels in cyclists, basketball and volleyball players at the end of T1 and T2.

**Figure 5 biology-11-00343-f005:**
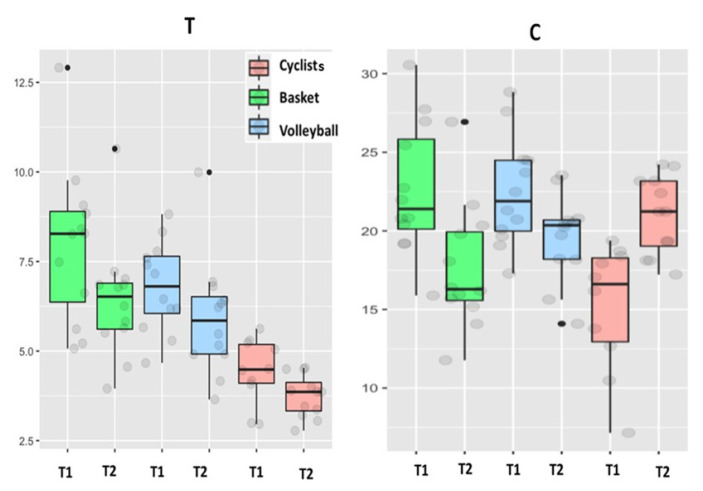
Testosterone (T) (ng/mL) and cortisol (C) (µg/dL) levels in cyclists, basketball and volleyball players at the end of T1 and T2.

**Figure 6 biology-11-00343-f006:**
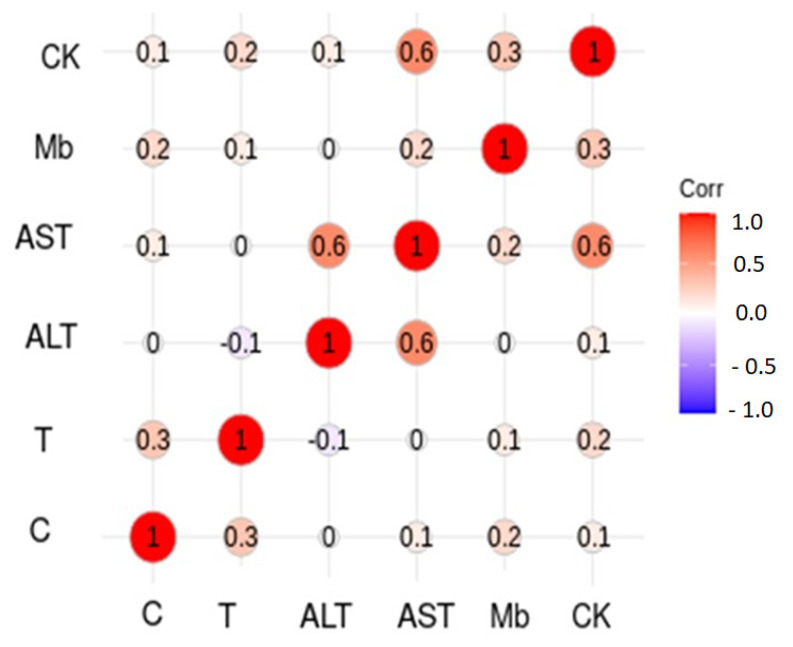
Correlation chart for muscular blood markers and cortisol and testosterone. Mild correlation = 0.3–0.5 and strong correlation > 0.6. Abbreviations used: AST, aspartate aminotransferase; ALT, alanine aminotransferase; C, cortisol; creatine kinase; Mb, myoglobin; T, testosterone; CK, creatine kinase.

**Table 1 biology-11-00343-t001:** Baseline parameters of participants in the study.

Sport	Cycling (*n* = 18)	Basket-Ball (*n* = 12)	Volley-Ball (*n* = 14)
Age (years)	26.2 ± 1.8	25.3 ± 4.4	25.7 ± 2.1
	(23.7–31.4)	(18.0–32.3)	(21.0–29.7)
Height (cm)	176.2 ± 3.4	198.0 ± 9.9	189.0 ± 0.1
	(168.5–183.4)	(181.2–214.5)	(188.8–189.1)
Weight (kg)	66.1 ± 3.6	96.8 ± 13.0	87.2 ± 4.1
	(56.4–71.5)	(75.4–121.6)	(79.8–94.5)
VO_2_ max (mL kg^−1^ min^−1^)	83.6 ± 2.7	56.5 ± 7.7	65.3 ± 4.2
	(75.5–91.5)	(38.2–78.2)	(60.4–76.3)

**Table 2 biology-11-00343-t002:** Changes in plasmatic parameters in cyclists, basketball and volleyball players at the end of T1 and at the end of T2.

	Cyclists		Basketball Players		Volleyball Players		Two-Way Anova
	T1	T2	T1	T2	T1	T2	
**TPP** (g/dL)	6.7 ± 0.4	6.2 ± 0.5 *	7.3 ± 0.6 ^†^	7.3 ± 0.5 ^‡^	7.5 ± 0.6 ^†^	7.6 ± 0.6 ^‡^	S, T×S
	(5.9–7.1)	(5.4–7.1)	(6.7–7.9)	(6.7–7.7)	(7.0–7.9)	(7.1–8.1)	
**Urea** (mg/dL)	40.7 ± 5.1	45.8 ± 4.9	41.8 ± 5.2	43.8 ± 5.7	41.7± 5.2	44.8 ± 5.0	
	(31–48)	(31–63)	(28–59)	(27–60)	(30–55)	(27–63)	
**Creatinine** (mg/dL)	0.8 ± 0.1	1.0 ± 0.1	1.2 ± 0.1 ^†^	1.2 ± 0.1 ^‡^	1.2 ± 0.1 ^†^	1.3 ± 0.1 ^‡^	S
	(0.6–1.0)	(0.7–1.1)	(1.1–1.5)	(1.1–1.4)	(1.0–1.4)	(1.2–1.7)	
**LDH** (U/L)	122 ± 19	359 ± 29 *	375 ± 31 ^†^	383 ± 33 ^‡^	349 ± 38 ^†^	330 ± 32	T×S
	(86–162)	(239–472)	(308–506)	(300–526)	(281–457)	(266–396)	

TPP: total plasmatic proteins, LDH: Lactate dehydrogenase. Data are shown as a 95% confidence interval for the mean + SD with normalization using bootstrap. Differences between groups were determined by independent *t*-tests (T1 vs. T2). Abbreviations used: LDH, lactate dehydrogenase; TPP, total plasmatic proteins. Significant differences (*p* < 0.05): * differences into each group (T1 vs. T2), ^†^ basketball and volleyball players vs. cyclists at T1, and ^‡^ basketball and volleyball players vs. cyclists at T2. T indicates a significant effect of training/competition; S indicates a significant effect of sport type; T×S indicates a significant interaction between T and S factors in a two-way ANOVA analysis.

**Table 3 biology-11-00343-t003:** Quantitative analysis of the changes observed in muscular damage blood markers in cyclists, basketball, and volleyball players at the end of T1 and at the end of T2.

	Cycling		Basketball		Volleyball		Two-Way Anova
	T1	T2	T1	T2	T1	T2	
CK (IU/L)	125 + 22 (94–178)	221 + 83 * (73–511)	226 + 123 ^#^ (197–443)	546 + 219 * (201–227)	220 + 50 (113–382)	243 + 72 (100–466)	T×S
Mb (ng/mL)	19.5 + 3.5 (13.0–26.0)	42.1 + 4.8 * (35.1–61.4)	35.4 + 5.4 ^†^ (22.1–48.4)	41.1 + 2.7 (35.6–49.1)	31.4 + 3.6 ^†^ (22.6–40.5)	34.75 + 4.5 (23.2–44.2)	T
AST (IU/L)	28 + 9 (16–60)	35 + 8 * (17–56)	31 + 5 (21–41)	35 + 7 (22–61)	24 + 3 ^†^ (17–30)	26 + 3 ^‡^ (17–33)	T×S
ALT (IU/L)	30 + 19 (13–101)	27 + 5 (12–36)	27 + 3 (20–36)	25 + 3 (17–35)	22 + 6 (15–42)	21 + 3 (15–28)	
Testosterone (ng/mL)	4.4 + 0.7 (3.0–5.7)	3.7 + 0.4 * (2.8–4.5)	7.9 + 1.4 ^†^ (5.1–12.9)	6.4 + 1.1 (4.0–10.6)	6.8 + 0.8 ^†^ (4.7–8.8)	5.9 + 1.0 * (3.7–10.0)	T, S
Cortisol (µg/dL)	15.2 + 2.9 (7.2–19.4)	21 + 1.6 * (17.2–24.2)	22.7 + 2.7 ^†^ (15.9–30.6)	17.7 + 2.6 * (11.8–26.9)	22.5 + 2.2 ^†^ (17.3–28.8)	19.7 + 1.8 (14.1–23.5)	T×S

Data are expressed as mean ± standard deviation (95% confidence interval). Differences between groups were determined by independent *t*-tests (T1 vs. T2). Abbreviations used: ALT, alanine aminotransferase; AST, aspartate aminotransferase; CK, creatine kinase; Mb, myoglobin. Significant differences (*p* < 0.05): * differences into each sport discipline, ^†^ basketball and volleyball players vs. cyclists at T1, and ^‡^ basketball and volleyball players vs. cyclists at T2. T indicates a significant effect of training/competition; ^#^ indicates significant differences between basketball players and cyclist at T1; S indicates a significant effect of sport type; T×S indicates a significant interaction between T and S factors in a two-way ANOVA analysis.

**Table 4 biology-11-00343-t004:** Differences in muscular damage blood markers comparing T1 vs. T2 in cyclists, basketball and volleyball players.

	Ciclyng	Basketball	Volleyball
	Δ (%)	Δ (%)	Δ (%)
**CK**	100	81.6	11
**Mb**	20	15.8	12.9
**AST**	23.6	12.9	83
**ALT**	−12	−8.1	−0.2

Difference in % between T1 vs. T2. Positive values indicate an increase and the negative ones a decrease. Abbreviations used: ALT, alanine aminotransferase; AST, aspartate aminotransferase; CK, creatine kinase; Mb, myoglobin.

## Data Availability

Data are available upon reasonable request.

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
