# Peer review of "Effects of Eccentric vs. Concentric Sports on Blood Muscular Damage Markers in Male Professional Players"

_biology, 2022, doi:10.3390/biology11030343_

Round 1

Reviewer 1 Report

The reviewer thank the authors for taking such an important subject for study. however the study was well designed i think that some changes should occur that may help making better its form. like the discussion section and the introduction.

Author Response

The reviewer thank the authors for taking such an important subject for study. However the study was well designed, I think that some changes should occur that may help making better its form, like the discussion section and the introduction.

Answer: We appreciate the comments raised by the reviewer to direct more precisely the contents of the manuscript. We hope that, the corrected version will fulfil all missing points and mistakes found.

 -In the abstract the results should appear more clearly with different tested levels.

Answer: We have changed the “Results” in the ABSTRACT accordingly.

-P2, L 62-64: Authors should more precise what is positive and what is considered negative, when they say "This is characterized by an increased presence of muscle proteins, such as creatine kinase (CK), together with lactate dehydrogenase (LDH) and myoglobin" (Mb), in blood. " what is about myoglobin in this kind of effort?

Answer: Myoglobin is mentioned in line 64 as an indicator of muscle damage. Nevertheless, the main concern regarding the use of these proteins as indicators of muscle damage is the different half-life that they display in circulation which makes difficult to integrate the results. We did not want to insist in this point because this is not the final objective of this work. In this sentence, we want just to indicate that high circulating levels of CK, LDH and Mb are just indicative of muscle damage. Degree and duration need to be assessed considering half life of each particular marker, but again this is not the goal of this research.

-P2, L 81/ Add a reference after "...in serum"

Answer: Reference 4 has been cited (P2 L70).

-The introduction contains important things but it is too long. Authors should rewrite the introduction and make it shorter. Also a relation between the introduction and the study protocol should clearly appear to give more appearance to the study originality.

Answer: Introduction has been shortened accordingly. The originality of our study shows for the first time a comparison of muscular damage generated by predominant eccentric or concentric sport disciplines. The main part of studies uses to work under controlled laboratory conditions that do not mimic a real sport situation. We have indicated this point in P3, L 103-104 and 128-129.

-P4, L161: consider changing to: Participation in the training and competition periods.

Answer: Subheading has been changed accordingly (P3, L 141).

-P4 L167: Is the moment of the season in which participants performed controlled exercises at lower intensity was the same for all three groups?

Answer: In this period, participants were working resistance routines: upper and lower part of the body for basketball and volleyball. Cyclist were working only the lower part and combined with road cycling at 60% VO2max. All participants started at low intensity but gradually increased performing routines at 80-90% of RM (repetition maximum) of each resistance exercise. In addition, cyclists trained climbing in mountain roads. These details are now included in the manuscript (P4, L 150-163), according to Reviewer suggestion. 

-Give more precision about the competition period. make a figure if necessary.

Answer: The competition period for basketball and volleyball consisted in 2 matches (one in the European league and the other in the National one). Recovery and resistance routines were performed the rest of the days. The competition period for cyclist was a Big Race (“Vuelta a España”) with very demanding stages (time trial, mountain climbing) and 2 days for active recovery and resting. These details are now indicated in the new version of the manuscript (P4 L 164-175).

-P4, L 175: "The training plan during T1 for basketball and volleyball was quite similar between them but different to cycling." in what kind of thing was similar between them? (intensity, duration, total of calories expended......? Please be more accurate.

Answer: Details of the training plans for basketball and volleyball players are indicated in the new version of the manuscript (P4, L 150-163).

-In figure 1 consisting of the training period, it is obvious to see the national competition planned in the same week with the international/european competition? authors must provide more explanations?

Answer: Yes, European competition occurs the same week than the National one. The European match is played at the middle of the week (Wednesdays) and the match of the National league is played during the week-end. This schedule has been organized by the European Federations. At the end, the best teams are classified for the Final Four. You can follow all matches at the TV and verify this particular point. Therefore, Figure 1 reflects this organization of the competition for basket and volleyball. A short explanation is provided in the text of the Figure (P4, L 179-181).

-P5, L214: "To control de correct " must be "To control the correct "

Answer: Typo has been corrected (P5, L 206).

-L522: "Intense and sustained exercises compromise muscle integrity favoring the release to circulation of muscle proteins (CK, Mb and LDH), limiting performance. " provide a reference here.

Answer: Reference 4 has been cited (P16,  L 515).

-L528; put a space between "integrity" and of

Answer: The space has been added.

-L541: provide a reference here.

Answer: Reference 5 has been cited (P16 L 533).

-L 559: "due the optimal recovery periods " add to after due and before the.

Answer: Recovery periods are after exercise performance. This has been clarified accordingly (P17 L 551).

-L 574-575: Authors are not allowed to give arguments for variables that were not studied in their study, like judging that "The changes that we observed in cortisol and testosterone in the  present study can be related to accumulated stress together with an optimal tissue regeneration. " otherwise they should provide a reference.

Answer: This is our interpretation. We have changed the sense of the sentence and indicated that this is our hypothesis for future research. In addition, this is a topic of research for other laboratories. See P17 L 564-571.

-My idea, since many limiting factors were cited for this study the discussion section should be wrote in another style taking into consideration those limit points when the authors want to confirm the results collected.

Answer: We propose always an interpretation for results obtained taking always into account the limiting factors. This is why the last paragraph contains a list of all limiting factors of the study. We need to show these limitations to the scientific community and encourage all readers to advance doing interventions with a design that can avoid these limitations. This is our Ethics in research. For these reasons, at the end of these paragraph we have indicated that these points need to be addressed in future researches (P17 L594-595).

Reviewer 2 Report

The manuscript I think deals with an interesting topic whose objective was "to analyze the pattern of blood markers of muscle injury in professional players at 2 points in the season: after a training and after a period of competition". After reading the manuscript certain questions and suggestions for improvement arise for me:

  1. In the introduction I have not seen any reference to the years 2020, 2021, 2022, please perform a thorough review of the subject and also include the references of the last two years and one month.
  2. Regarding the participants it is not indicated how the sample calculation was performed, if it is a convenience sample, please include the statistical power achieved with the number of athletes included in each group.
  3. The statistical analysis includes the performance of a correlation, however, it is not indicated how the result of the correlation will be interpreted. In the results section, it is mentioned that there is a strong correlation, medium correlation, etc. However, I have not found within the manuscript the description of how the correlation coefficient is to be interpreted. Please indicate how the correlation coefficient will be interpreted.
  4. I believe that the discussion would be improved by including a paragraph explaining clearly and concisely the implications of the study.
  5. In the conclusion, I suggest not using abbreviations. I think it is not appropriate to include this sentence in the conclusion "In this context, we believe that immunomodulatory protocols (supplements and specific training routines) could be instrumental in improving recovery by reducing muscular damage [14,15,37,38]" since in this research, as far as I have understood, the role of supplements was not evaluated, and I think it is neither usual nor appropriate to use quotations in the conclusion, I think this is appropriate for the discussion section.

Author Response

The manuscript I think deals with an interesting topic whose objective was "to analyze the pattern of blood markers of muscle injury in professional players at 2 points in the season: after a training and after a period of competition". After reading the manuscript certain questions and suggestions for improvement arise for me:

  1. In the introduction I have not seen any reference to the years 2020, 2021, 2022, please perform a thorough review of the subject and also include the references of the last two years and one month.

Answer: Reviewer is right. References 4 and 5 are from 1997 and 1999. We have changed for more actual references from 2019 and 2021.

  1. Regarding the participants it is not indicated how the sample calculation was performed, if it is a convenience sample, please include the statistical power achieved with the number of athletes included in each group.

Answer: It is difficult to find participants of such as very high level. For this reason this is a convenience sample. We have indicated this point in Materials and Methods (P3, L 147-149).

  1. The statistical analysis includes the performance of a correlation, however, it is not indicated how the result of the correlation will be interpreted. In the results section, it is mentioned that there is a strong correlation, medium correlation, etc. However, I have not found within the manuscript the description of how the correlation coefficient is to be interpreted. Please indicate how the correlation coefficient will be interpreted.

Answer: Figure 6 shows the correlations between the blood markers of muscle damage and cortisol and testosterone. A strong correlation exists between CK and AST. However, cortisol and testosterone only have a mild correlation between them, with no correlation to the rest of protein blood markers. Finally, a strong positive correlation exists between AST and ALT. The color in the correlation chart (Figure 6) indicates the level of correlation: stronger color corresponds to high correlation. The correlation scale is in the upper right of Figure 6.

  1. I believe that the discussion would be improved by including a paragraph explaining clearly and concisely the implications of the study.

Answer: These data are valuable to the broader scientific community or other athletes and coaches from comparable sports. At the end of Discussion we have included implications and limitations of our study (P17 L 572-575).

  1. In the conclusion, I suggest not using abbreviations. I think it is not appropriate to include this sentence in the conclusion "In this context, we believe that immunomodulatory protocols (supplements and specific training routines) could be instrumental in improving recovery by reducing muscular damage [14,15,37,38]" since in this research, as far as I have understood, the role of supplements was not evaluated, and I think it is neither usual nor appropriate to use quotations in the conclusion, I think this is appropriate for the discussion section.

Answer: Abbreviations are not used in the Conclusion as suggested by the Reviewer. In addition, we have eliminated the sentence indicated by the Reviewer because is not in the aim of the research (P18 L 600-601).

Round 2

Reviewer 2 Report

Thank you for your response.

In order for the manuscript to be accepted the authors must include what is the statistical power of the sample included in their study. This is very relevant in order to determine the validity of the study.

Please reference which classification you have followed to categorize the correlations as medium or strong, it is important that you reference the manuscript you have used to make that interpretation.

Author Response

Thank you for your response.

In order for the manuscript to be accepted the authors must include what is the statistical power of the sample included in their study. This is very relevant in order to determine the validity of the study.

Answer: In the present study, we have worked with a sample size from 12 to 18 individuals. In all variables studied, we obtain a Cohen's d value greater than 0.9 for n=18 and 0.7 for N=12 and 14. This implies that for our significance level 0.05 our sample size has a power greater than 0.8 (meaning by "power" the probability that the test correctly rejects the Null Hypothesis if the Alternative Hypothesis is true). This statement is reflected in section 2.5 (P6, L 253-257.

Please reference which classification you have followed to categorize the correlations as medium or strong, it is important that you reference the manuscript you have used to make that interpretation.

Answer: In this study by "mild" correlation we mean close to 0.3-0.5, by "strong" correlation we mean more than 0.6. This is explained in the legend of Figure 6 (P 14, L 440-441).